# Exploring the Immunomodulatory Aspect of Mesenchymal Stem Cells for Treatment of Severe Coronavirus Disease 19

**DOI:** 10.3390/cells11142175

**Published:** 2022-07-12

**Authors:** Jitendra Kumar Chaudhary, Deepika Saini, Pankaj Kumar Chaudhary, Anurag Maurya, Ganesh Kumar Verma, Akhilesh Kumar Gupta, Rakesh Roshan, Tarun Kumar Vats, Nidhi Garg, Deepika Yadav, Nimita Kant, Anil Kumar Meena, Anissa Atif Mirza-Shariff

**Affiliations:** 1School of Life Sciences, Jawaharlal Nehru University, New Delhi 110067, India; 2Department of Zoology, Shivaji College, University of Delhi, New Delhi 110027, India; rakesh.biotech85@gmail.com (R.R.); tarunkumarvats@shivaji.du.ac.in (T.K.V.); nidhigarg1804@gmail.com (N.G.); dy_277@yahoo.com (D.Y.); nimita@shivaji.du.ac.in (N.K.); 3Department of Biochemistry, All India Institute of Medical Sciences (AIIMS), Rishikesh 249201, India; deepikadeepasaini@gmail.com (D.S.); ganeshkumarverma1992@gmail.com (G.K.V.); 4Molecular Biology & Proteomics Laboratory, Department of Biotechnology, Indian Institute of Technology (IIT), Roorkee 247667, India; pkchaudharydu@gmail.com; 5Department of Botany, Shivaji College, University of Delhi, New Delhi 110027, India; anuragvns.maurya@gmail.com; 6School of Medical Science and Technology (SMST), Indian Institute of Technology (IIT), Kharagpur 721302, India; akhileshgobi97@gmail.com; 7Department of Zoology, Maharshi Dayanand Saraswati University, Ajmer 304001, India; anilmeena.01987@gmail.com

**Keywords:** Severe Acute Respiratory Syndrome Coronavirus-2, angiotensin converting enzyme 2, cytokine storm, mesenchymal stem cells, immunomodulation

## Abstract

Severe Acute Respiratory Syndrome Coronavirus 2 (SARS-CoV-2) is an enveloped, positive sense, single stranded RNA (+ssRNA) virus, belonging to the genus Betacoronavirus and family Coronaviridae. It is primarily transmitted from infected persons to healthy ones through inhalation of virus-laden respiratory droplets. After an average incubation period of 2–14 days, the majority of infected individuals remain asymptomatic and/or mildly symptomatic, whereas the remaining individuals manifest a myriad of clinical symptoms, including fever, sore throat, dry cough, fatigue, chest pain, and breathlessness. SARS-CoV-2 exploits the angiotensin converting enzyme 2 (ACE-2) receptor for cellular invasion, and lungs are amongst the most adversely affected organs in the body. Thereupon, immune responses are elicited, which may devolve into a cytokine storm characterized by enhanced secretion of multitude of inflammatory cytokines/chemokines and growth factors, such as interleukin (IL)-2, IL-6, IL-7, IL-8, IL-9, tumor necrosis factor alpha (TNF-α), granulocyte colony-stimulating factor (GCSF), basic fibroblast growth factor 2 (bFGF2), monocyte chemotactic protein-1 (MCP1), interferon-inducible protein 10 (IP10), macrophage inflammatory protein 1A (MIP1A), platelet-derived growth factor subunit B (PDGFB), and vascular endothelial factor (VEGF)-A. The systemic persistence of inflammatory molecules causes widespread histological injury, leading to functional deterioration of the infected organ(s). Although multiple treatment modalities with varying effectiveness are being employed, nevertheless, there is no curative COVID-19 therapy available to date. In this regard, one plausible supportive therapeutic modality may involve administration of mesenchymal stem cells (MSCs) and/or MSC-derived bioactive factors-based secretome to critically ill COVID-19 patients with the intention of accomplishing better clinical outcome owing to their empirically established beneficial effects. MSCs are well established adult stem cells (ASCs) with respect to their immunomodulatory, anti-inflammatory, anti-oxidative, anti-apoptotic, pro-angiogenic, and pro-regenerative properties. The immunomodulatory capabilities of MSCs are not constitutive but rather are highly dependent on a holistic niche. Following intravenous infusion, MSCs are known to undergo considerable histological trapping in the lungs and, therefore, become well positioned to directly engage with lung infiltrating immune cells, and thereby mitigate excessive inflammation and reverse/regenerate damaged alveolar epithelial cells and associated tissue post SARS-CoV-2 infection. Considering the myriad of abovementioned biologically beneficial properties and emerging translational insights, MSCs may be used as potential supportive therapy to counteract cytokine storms and reduce disease severity, thereby facilitating speedy recovery and health restoration.

## 1. Introduction

Mesenchymal stem cells (MSCs), one of the prominent categories of adult stem cells (ASCs), are heterogenous, multipotent progenitor cells capable of delaying aging and restoring organ homeostasis following occurrence of inflammation, injury, and disease [1,2,3]. MSCs are primarily located in specialized niches across the multiple vascularized tissues and/or organs, such as bone marrow, adipose tissue, Wharton’s jelly, umbilical cord, dental pulp, and the placenta [4,5,6,7], although it is originally isolated from the bone marrow of the mammal [8]. Following isolation and characterization, MSCs are expanded through in vitro culture for empirical research aimed at finding their fundamental biology and potential therapeutic applications. MSCs show capabilities for in vitro tri-lineage differentiation, such as cartilage, bone, and adipocyte [9]—one amongst three important criteria for characterization and validation of their stemness as set forth by International Society for Cellular Therapy (ISCT). In particular, neuronal differentiation of MSCs—a type of ectodermal differentiation—is gaining much attention owing to its applicability for potential treatment of neurodegenerative diseases [10,11,12]. Besides, they show plastic adherence, and display positive expression of cell surface markers CD29, CD73, CD90, and CD105, while being devoid of CD11b, CD14, CD34, CD45, CD79, and HLA (Human leukocyte Antigen)-DR surface markers [13,14]. Nonetheless, the surface expression of MSC-positive markers is highly variable and predominantly depends on their source of origin and niche. Generally, MSCs are intricately involved in and contribute towards maintenance and remodeling of histological architecture of various organs. They perform a myriad of functions through complex cellular (cell-to-cell) interaction and molecular mechanisms of action, thereby facilitating multilineage differentiation and immunomodulation, as well as playing crucial reparative roles in bone diseases, degenerative disorders, myocardial infarction, burns and chronic wounds, spinal cord injury, neurodegeneration, autoimmune diseases, and inflammatory diseases, among others [15,16,17]. Owing to stromal cell-derived factor-1 (SDF-1)/CXC chemokine receptor 4 (CXCR4) axis-guided homing properties [18] and secretion of multitude of bioactive factors of medicinal importance, MSCs are capable of inducing tissue-specific resident stem cells to repair and/or remodel tissues and are, therefore, also appropriately referred to as Medicinal Signaling Cells [14]. The therapeutic effects of MSCs are primarily mediated via secretion of multitude of soluble bioactive molecules comprising cytokines, chemokines, growth factors, angiogenic factors, as well as reparative peptides/proteins-, mRNA-, and microRNA-containing extracellular vesicles (MSC-EVs), which are collectively referred to as “MSC-secretome”. In fact, there have been reports of considerable improvement in disease conditions involving injured animal models, including damaged cartilage and bone, myocardial, hepatic, and neural tissues, following allogenic transplantation of MSCs [19,20,21], highlighting the broad-spectrum therapeutic potential of stem cell. For instance, Zhao et al. recently reported the attenuation of mtDNA damage and inflammation in an acute kidney injury mouse model. This therapeutic outcome following intravenous injection of EVs (extracellular vesicles) was found to be potentially mediated via transfer of mitochondrial transcription factor A (TFAM) mRNA, thereby restoring TFAM protein and TFAM-mtDNA, resulting in substantial improvement in mtDNA, mitochondrial oxidative phosphorylation (OXPHOS), and inflammation [22]. This finding offers promising nanotherapeutics for myriad of diseases characterized by mitochondrial damage and inflammation, such as COVID-19. MSCs also secrete growth factors, such as keratinocyte growth factor (KGF), vascular-endothelial growth factor (VEGF), hepatocyte growth factor (HGF), and angiopoietin-1 (Ang-1), and thereby play crucial role in the repair of the alveolar epithelial and endothelial cells [23]. In addition, MSCs engage with immune cells via secretion of a range of factors, such as TGF-β, IL-1RA, IL-6, IL-10, indole 2,3 dioxygenase (IDO), prostaglandin E2 (PGE2), and nitric acid, thereby accomplishing paracrine immunomodulation [24,25]. In fact, anti-inflammatory cytokine IL-10 plays an important role as an anti-fibrotic molecule and, therefore, can potentially be used to modulate chronic condition/disease in various organs, including heart and lungs [26]. MSCs’ anti-fibrotic property is mainly attributed to a decrease in collagen secretion/deposition and a down regulation of matrix-metalloproteinase [27]. As pulmonary fibrosis is also distinctly reported in severe COVID-19 patients [28], IL-10 supplementation along with other therapy (including MSC-based) may be quite effective. Moreover, direct cell-to-cell contact/communication helps in MSC-mediated immunosuppressive effect, as well as capable of transferring mitochondria to injured cells through formation of tunneling nanotubes (TNT) [29], thereby restoring the cell viability prerequisite for reinvigorating vital cell/tissue functions, including energy metabolism, stress response, cell growth, and apoptosis. Among the several underlying mechanisms of immunosuppression by MSCs, negative regulation of T and B cells proliferation and functions via engagement of the programmed death molecule (PD-1) with its ligands (PG-L1, PG-L2), upregulation of T regulatory cell (Treg) function, inhibition of natural killer (NK) cell activity, and PGE2-mediated prevention of dendritic cell (DC) maturation and functions, are quite frequently reported [25,30,31]. MSC-mediated inhibition of B cell functions, including proliferation, plasma cell differentiation, and antibody production, are largely dependent on both CD4^+^ and CD8^+^ T cells [32]. Besides, MSCs release bioactive factors, which may act as a potential antiviral, antibacterial, and analgesic [33,34], expanding stem cells’ horizon of therapeutic applications. However, these immunomodulatory and reparative biological properties of MSCs are not constitutive in nature and may be greatly influenced by the existing micro-environment’s inflammatory conditions and associated factors (Figure 1).

Taxonomically and phylogenetically, Severe Acute Respiratory Syndrome Coronavirus 2 (SARS-CoV-2) is a member of the family *Coronaviridae* and subfamily *Coronavirinae* which, in turn, consists of following four genera; namely, *Alphacoronavirus*, *Betacoronavirus*, *Gammacoronavirus,* and *Deltacoronavirus*. Structurally, SARS-CoV-2 is a fully enveloped virus, bearing spherical morphology with diameter ranging from 100–160 nm. It possesses a positive single-stranded RNA molecule of around 30 kilobases (kb) as its genetic material and four structural proteins; namely, spike (S), envelope (E), membrane (M), and nucleocapsid (N), apart from several enzymatic proteins and accessory factors required for progeny formation [35]. Coronavirus is primarily transmitted from an infected person to healthy individuals through nasal inhalation of virus-laden respiratory droplets, causing COVID-19 (coronavirus disease 19). The pathogenesis of SARS-CoV-2 involves molecular interaction between virus heterotrimeric spike (S) protein (one of the four categories of structural proteins) and ACE2 receptor on the host cell surface, which is primed by cellular transmembrane protease, serine 2 (TMPRSS2, also known as spike protein activator), facilitating the virus’s cellular entry [36,37]. The ACE-2 receptor, a crucial determinant of infection, is highly expressed on various types of cells, including adipocytes, epithelial cell lining of respiratory and gastrointestinal tracts, as well as on some immune cells (monocyte and macrophage) [38], making such receptor-bearing cells highly susceptible to SARS-CoV-2 entry and subsequent infection. Surprisingly, certain SARS-CoV-2 variants, including the E484D S variant, may not require an ACE-2 receptor to infect cells, as evidenced from the study involving human H522 lung adenocarcinoma cells [39], highlighting the existence of alternative pathway(s) of virus entry that may be further explored for therapeutic targeting to counteract and mitigate multiorgan infection and, consequently, check disease spread. As of 5:40pm CEST, 22 June 2022, the World Health Organization (WHO) has reported over 538,321,874 confirmed coronavirus cases and 6,320,599 deaths globally (https://covid19.who.int/, accessed on 23 June 2022) since its outbreak in December 2019 in Wuhan, Hubei Province, China. Although the majority of infected persons are asymptomatic and/or mildly symptomatic, critically ill patients manifest pathological symptoms, such as ground glass opacity in lungs, decreased lymphocytes (lymphocytopenia), increased neutrophils (neutrophilia/neutrophilic leukocytosis), and cytokine storms owing to higher concentration of inflammatory cytokines; namely, TGF-α, TGF-β, IP10, MIP1A, IL-1RA, IL-6, GCSF, MCP-1, and indole 2,3 dioxygenase (IDO) [40,41]. Following the localized release of these cytokines/chemokines and associated factors, there is an occurrence of enhanced production of reactive oxygen species (ROS), increased vascular permeability, compromised lung barrier function, enhanced secretion of alveolar proteinaceous exudate, and pulmonary fibrosis [42]. Besides, persistent cytokine storm causes considerable tissue and/or organ injury, including alveolar damage, hypoxemia, pulmonary oedema, and progressive respiratory failure, clinically referred as acute respiratory distress syndrome (ARDS), which may culminate in systemic inflammatory response syndrome (SIRS), leading to the death of patients [43,44]. Although the respiratory failure is the primary reason for most of the fatality following SARS-CoV-2 infection, there are considerable deteriorative changes in heart and liver functions as well, compounding the existing clinical condition, thereby contributing towards enhanced mortality as a secondary consequence [45]. 

Among the multiple treatment modalities available—including glucocorticoid therapy, convalescent plasma therapy, monoclonal antibody, anti-interleukin (IL-6) receptor antibody therapy, and lipid raft targeting [46,47], none of them has proven to be curatively successful against coronavirus disease 19, compelling clinical researchers to look for effective novel treatment regimes. In this regard, mesenchymal stem cells (MSCs), owing to their immunomodulatory, anti-inflammatory, anti-oxidative, anti-apoptotic, pro-angiogenic, pro-regenerative, anti-fibrogenic, and reparative properties as observed in multiple animal model studies and in vitro human lung models, may be one of the promising therapeutic avenues for the treatment of COVID-19 [48,49,50]. Furthermore, MSCs are immune-evasive owing to negligible expression of MHC-I, complete lack of MHC-II and co-stimulatory molecules B7-1, B7-2, or CD40 [51], which may enhance their therapeutic compatibility and applications without eliciting immunogenicity and/or antigenicity to the dangerous level (no adverse immune reaction) during and/or post allogeneic transplantation. The absence of noticeable/detectable adverse immune reactions, coupled with regenerative and immunomodulatory properties of MSCs following transfusion in severe COVID-19 patients, is very strong and unequivocal proof-of-concept regarding immunotherapeutic suitability of MSC therapy. Here, we discuss the typical characteristics of MSCs, including mechanisms underlying MSC-induced immunomodulation, pathological consequence of over-activation of both innate and adaptive immune systems, and relevant clinical trials with MSCs for supportive and curative therapeutic effect-based treatment modalities following SARS-CoV-2 infection, especially in severe and critically ill COVID-19 patients.

## 2. Immune Responses to SARS-CoV-2 Infection

SARS-CoV-2 virion consists of a single RNA molecule, several accessory factors, and four categories of structural proteins: namely, spike (S), membrane (M), envelope (E), and nucleocapsid (N). These structural proteins intrinsically possess differential levels of immunogenicity [52], thereby eliciting a varied extent of immune responses with considerably different clinical consequences. Therefore, such structural proteins, especially glycosylated trimeric S protein [53] owing to its prominent immunogenic nature and crucial determinant of infection, are also the target for therapeutic intervention, apart from several host-derived cellular factors [54]. The term “cytokine storm”, originally coined to describe a graft-vs-host disease (GvHD) [55], broadly encompasses unprecedented release of cytokines and/or chemokines, including TGF-α, TGF-β, IP10, MIP1A, IL-1RA, IL-6, GCSF, and MCP-1, by immune cells following onset of infectious and autoimmune diseases, sepsis, cancer, haemophagocytic lymphohistiocytosis, and acute immunotherapy response [56,57,58]. This causes a range of clinical manifestation, including increased local temperature, rash, nausea, arthralgia, and depression, with substantial local and systemic detrimental effects on multiple organs, leading to multi-organ failure. For instance, there is diffuse alveolar damage accompanied by infiltration of interstitial lymphocytes and hyaline membrane formation as a consequence of cytokine outburst [40]. Although, it still remains unclear how the host’s response to infection triggers inflammatory sequence of events, resulting in cytokine storm; nonetheless, it is believed to be caused by increased immune cell activation via Toll-Like Receptor (TLR)-mediated signaling, on the one hand, and reduced anti-inflammatory response, on the other. In addition to cytokine storm, COVID-19 patients with severe-to-critical symptoms show elevations in levels of D-dimer, C-reactive protein (CRP), procalcitonin, and ferritin, along with substantial reduction in lymphocytes (B and T cells) and NK cell counts [59]. Therefore, developing insight into the underlying kinetics of the abovementioned processes, involving both innate and adaptive immune systems, along with understanding other relevant clinical parameters, may be used as surrogate biomarkers to predict and monitor the development of COVID-19. Such holistic understanding can be appropriately employed and tailored to patient-specific clinical requirements. 

### 2.1. Innate Immune Response Following SARS-CoV-2 Infection/COVID-19

The innate immune system acts as non-specific, frontline host defense, comprising of dendritic cells, monocyte, macrophage, neutrophils, circulating monocytes, granulocytes, NK cell, and even some non-immune cells, such as endothelial cells (ECs) and smooth muscle cells with highly adaptable immunological functions, which collectively render protection against danger signals. NK cells are also referred to as innate lymphoid cells (ILCs) but, unlike conventional lymphoid B and T cells, they do not undergo expression of rearranged antigen receptors, such as BCRs (B Cell Receptors) on B cells and TCRs (T Cell Receptors) on T cells [60]. Infectious SARS-CoV-2, carried through virus-laden droplets, primarily targets airway epithelial cells, wherein they enter via the high affinity ACE-2 receptor, and start replicating and spreading downstream of airways, leading to progression towards COVID-19. The ACE-2 receptor is highly expressed on various types of cells, including adipocytes, epithelial cells of the respiratory tract and intestine, as well as immune cells like monocyte and macrophage, making these cells highly susceptible to SARS-CoV-2 infection. Following infection, the expression level of ACE2 on peripheral blood (PB) monocytes downregulates, perhaps as a function of secondary outcome to viral attachment and subsequent cellular signaling. COVID-19 patients show normal level of monocytes with activated phenotype, as evidenced by high FSC (forward side scatter) and potential to secrete cytokines, such as IL-6, IL-10, and TNF. The level of such activated monocytes corresponds to disease severity and poor prognosis, and can be characterized by immunophenotyping involving panoply of surface markers, such as CD11b, CD14, CD16, CD68, CD80, CD163, and CD206 [38]. Besides, PB monocytes with positive expression of CD163 and CD206 are regulatory in nature that may impact antiviral effector T cell response. In addition to immune effects, CD163 positive monocytes have also been associated with hemophagocytic lymphohistiocytosis syndrome [61] which, in turn, may be attributed to cytokine outburst and consequent immunologic pathology observed in COVID-19 patients [62]. Similarly, SARS-CoV-2 triggers release of IL-6, TNF, IL-10, and PD-1 from alveolar macrophage, which may contribute towards and compound ongoing cytokine storm, causing lymphocytopenia. COVID-19 patients with ARDS show accumulation of monocyte-derived inflammatory FCN1^+^ macrophages in their bronchoalveolar fluid (BALF) as revealed by single cell RNA sequencing. Moreover, transcriptional analysis of BALF and PB mononuclear cells confirmed substantially high level of chemokines, such as IFN-inducible protein 10 (IP-10) and monocyte-chemotactic protein-1 (MCP-1), among others, which might help macrophages in infiltrating the site of tissue and/or organ infection in COVID-19 patients, and this rationale is found to be consistent with autopsy reports as well [63]. Together, these inflammatory monocyte/macrophages may lead to type I IFN-mediated dysregulated inflammatory response, including apoptosis of T cells (causes lymphocytopenia), contributing to underlying pathophysiology [59,64]. There can be a spectrum of inflammatory responses, and in most cases immune response is quite capable of neutralizing the virus; however, compromised and/or weakened immune response may cause respiratory failure accompanied by increased capillary leakage in severe COVID-19 patients and eventual death. 

The abundance of neutrophils in circulation slowly increases as COVID-19 progresses; therefore, this elevated level of neutrophil may be useful for monitoring and predicting disease dynamics and severity. However, the evaluation of disease severity based on combination of IgG with a neutrophil-to-lymphocyte ratio (NLR) might be even a better predictor compared to neutrophil count alone [65,66]. Although, NETosis—the formation of extracellular webs of DNA/histones released by neutrophils, referred to as neutrophil extracellular traps (NETs)—controls infection, its aberrant production might contribute to cystic fibrosis, excessive venous and arterial thrombosis, and cytokine storm, which may culminate in ARDS [67,68,69]. Further, a high level of chemokines, such as CXCL-2 and CXCL-8 from PB mononuclear cells, may help infiltrate a considerable number of neutrophils to the site of infection, enhancing pulmonary inflammatory response. Therefore, NETosis may be a viable therapeutic target to mitigate the extent of severity and outcome of the SARS-CoV-2 pandemic.

Natural Killer (NK) cells, present in healthy lymphoid and mucosal tissues, provide a rapid and effective immune response against a multitude of pathogens and tumors, and are also involved in the maintenance of immune homeostasis. In general, immature NK cells are characterized as CD56^bright^ and secrete proinflammatory cytokines, such as IFN-γ, whereas mature NK cells are CD56^dim^KIR^+^ CD16^high^, carrying out cytotoxic function in response to the loss of human leukocyte antigen (HLA) class I [70]. The killer-immunoglobulin receptors (KIRs) are acquired along the CD16 (FcRγIIIA) during NK cell development and are a prerequisite for cytotoxic function. The cytotoxic effect of NK cells from COVID-19 patients has been found to be antagonized by reduced expression of CD107a, ksp37, granulysin, and granzyme B; such NK cells display impaired production of chemokines, IFN-γ, and TNF-α, among others [71]. There is a recent report of anti-spike dependent NK cell response during COVID-19 vaccine study in macaques [72], validating the NK cell’s functional involvement towards COVID-19-infection; however, such an immune response is quite variable and ambiguous, and hence requires deeper empirical insight. For instance, CD158b^+^ NK cells were found to positively correlate with the presence of anti-SARS-CoV-2 antibodies and disease severity. In contrast, a recent study reported an inverse relation between disease severity and the number of NK cells in peripheral blood; whereas another report showed no difference in the number of CD16^+^ CD56^+^ NK cells in mild vs. severe cases [73], requiring more empirical evidence to draw an unequivocal conclusion. Further, reduced NK cell frequency and CD16 expression reverted to normal levels following COVID-19 recovery [74]. Considering these facts, additional studies are needed to understand whether NK cells contribute to cytokine storm, and its involvement, if any, in virus clearance during the course of COVID-19.

Conventionally, the innate immune system primarily relies on pattern-recognition receptors (PRRs) to detect viral single-stranded RNA (ssRNA), such as SARS-CoV-2, whereas, cytosolic RIG-I like receptors (RLRs), and extracellular and endosomal Toll-like receptors (TLRS) are capable of sensing double stranded RNA (dsRNA) and eliciting an immune response. Following activation through a multitude of signaling pathways, innate immune cells secrete cytokines and chemokines, such as IFN-γ, TGF-α, TGF-β, IP10, MIP1A, IL-1RA, IL-6, IL-10, GCSF, MCP-1, CXCL-2, and CXCL-8, which naturally tend to neutralize the infectious virus, a process referred to as antiviral response. Among such anti-viral responses, type I/III interferons (IFNs) play most crucial role, even in the case of COVID-19, as some early evidence showed sensitivity of SARS-CoV-2 to IFN-I/III [75]. To counteract the anti-viral response, SARS-CoV-2 has evolved multiple inhibitory mechanisms to prevent IFN-I induced signaling. For instance, there is a report of a lack of robust type I/III IFN signatures from primary bronchial cells, infected cell lines, and a ferret model [76]. This study is further corroborated by impaired IFN-I signature in several ill COVID-19 patients, compared with mild and moderate cases [77]. Therefore, SARS-CoV-2 is relying on multiple mechanisms of evasion, including inhibition of crucial steps of the antiviral signaling pathways, from PRR sensing and cytokine secretion to highly conserved IFN signal transduction. Considering the abovementioned empirical evidences, a holistic therapeutic strategies must be designed to strike such susceptible molecular targets on virus, leading to effective and efficient virus eradication, neutralization, and consequent disease containment.

### 2.2. Adaptive Immune Response and COVID-19 

The adaptive immune system, comprised of T and B lymphocytes, is highly specific in terms of its ability to develop specific antibody-producing plasma cells and corresponding memory cells. At the later and subsequent stages of infection, memory cells help in eliciting precise immune response against previously encountered immunological danger, broadly referred to as antigen. This incredible immunological task is accomplished owing to the intrinsic potential of T- and B-lymphocytes to precisely undergo genomic rearrangement and generate unique antigen receptors, called T cell receptors (TCRs) and B cell receptors (BCRs), correspondingly [78]. This leads to generation of tremendous diversity at the level of TCRs and BCRs that are estimated to be of around 10^13^ distinct types, which is sufficient to recognize and engage with myriad of viral, bacterial, and various other categories of pathogen-associated antigens in an incredibly precise manner [79]. In general, T and B cells work in concerted fashion, wherein CD4^+^ T cells molecularly assist B cells via paracrine and/or juxtacrine signalling for production of antibodies, and synchronizes the response of other immune cells. CD8^+^ T cells cytotoxically target infected cells to eliminate viruses, causing tremendous reduction in viral loads. In contrast, dysregulated T and B cell responses result in immunopathology and enhanced disease severity rather than immunoprotection. Therefore, we discuss how T and B cells respond to SARS-CoV-2 alongside innate immune cells during the course of infection and, therefore, determine the disease prognosis and outcome. 

The functional effectiveness of T lymphocytes against infectious agents depends on their capability to recognize and mount a timely and appropriate immune response to HLA-complexed antigenic epitope displayed on nucleated cells and/or professional antigen presenting cells (pAPCs). Human leukocyte antigen (HLA) system includes groups of related proteins that are encoded by the major histocompatibility complex (MHC) genes. Although, MHC-I and MHC-II-complexed antigenic peptides are independently recognized by CD8^+^ T cells and CD4^+^ T cells, respectively, but their functional activation are quite interdependent and intricately synchronized. There have been several reports showing reduction in peripheral blood CD4^+^ and CD8^+^ T cells’ count in moderate and severe COVID-19 patients [80,81], which is quite similar to the earlier finding pertaining to SARS-CoV-1 infection [82]. Furthermore, the extent of inflammatory cytokine in serum, such as IL-6 and TNF-α, as well as lymphopenia, especially with respect to CD8^+^ T cells, positively correlates and serves as a better predictor for severity and mortality in COVID-19 [81]. Among the multiple mechanisms underlying the reduction of T cells in peripheral blood of severe COVID-19 cases, they may involve cytokine-induced retention of T cells in lymphoid organs and excessive attachment to endothelium, considerable death of lymphocyte owing to Fas-FasL interactions, and excessive cytokine-induced cell death [83], as supported by reports. For instance, there has been report of substantial infiltration of lungs by CD8^+^ T cell as revealed by single cell RNA-seq analysis of bronchoalveolar lavage (BAL) fluid drawn from COVID-19 patients [84]. Besides, postmortem examination of COVID-19 patients revealed extensive lymphocyte infiltration in the lungs [40]. However, another study involving post-mortem examination found only neutrophilic infiltration [45] and, therefore, the underlying cause of lymphopenia following viral infections, including SARS-CoV-2, remains varied and elusive, necessitating further in-depth study. The reasons for such differential findings may be related to disease severity, immunological state, and genetic background of patients, amongst others.

There are reports characterizing specific T cell immunity following SARS-CoV-2 infection. Examination of 12 recovering COVID-19 patients showed robust T cell response specific to viral M, N, and S structural proteins as revealed by IFN-γ ELISPOT (enzyme-linked immunospot) assay. In another study involving flow cytometry-based PBMCs analysis from moderately to severely ill COVID-19 patients, around 1.4% and 1.3% virus-specific CD4^+^ and CD8^+^ T cells, respectively, were detected in all patients post 15 days ICU admission. There have been findings of a preferential specificity of both CD4^+^ and CD8^+^ T cells for overlapping S protein’s epitopes accompanied by functional induction of IFN-γ, IL-2, and TNF-α, as well as lower levels of IL-5, IL-13, IL-9, IL-10, and IL-22, as characterized by ELISA. Another report focusing on S-specific CD4^+^ T cell following incubation with S protein-derived overlapping peptide pools showed induction of CD154 and CD137 co-expression. Together, these studies indicate induction of heightened adaptive immune response to combat SARS-CoV-2 [85,86,87]. Therefore, developing deeper insight into underlying mechanisms that elicit the adaptive immune system may be helpful in designing intervention and treat infection.

## 3. MSCs as Potential Therapeutic Cells to Mitigate Severity of COVID-19

Whereas the entire world is still eagerly looking for the suitable drug discovery/repurposing and invention of safe and effective treatment(s) for COVID-19, the disease is on the rampage, infecting lakhs and killing thousands of people each day. The underlying cause of thousands of daily deaths following SARS-CoV-2 infection is manifold, including cytokine storm-induced oedema, reduced airway exchange, and development of acute respiratory distress syndrome (ARDS), which culminate in multiorgan failure and eventual death [88]. The ARDS is typically characterized by increased permeability of pulmonary capillary endothelial cells, alveolar epithelial cells, infiltration by inflammatory cells, and lung fibrosis that is further compounded by adverse thromboembolic events and acute cardiac injury [89]. As per the current estimate, around 80% of SARS-CoV-2 infected persons show very mild to moderate symptoms, whereas 15% suffer from severe pneumonia, and the remaining 5% develop ARDS, which is very likely to culminate in septic shock and multiple organ system failure [40,44]. The mortality rate among critically ill COVID-19 patients (those developing ARDS) ranges from 34.9% to 46.1% [90]. Among multiple ongoing therapeutic interventions delivered on a trial basis, mesenchymal stem cells-induced immunomodulation may be realized as a supportive therapy in the light of emerging promising evidences as comprehensively reviewed by Levy et al. in recently published work [91]. MSCs are multipotent adult stem cells, capable of differentiating into ectodermal, mesodermal, and endodermal lineages in vitro [13] and located in specialized micro-anatomical niches, including the perivascular region of various tissues/and or organs, such as bone marrow (BM), adipose tissue (AT), umbilical cord (UC), umbilical cord blood (UCB), Wharton’s jelly (WJ), amniotic fluid, placenta (PL), dental pulp, periodontal ligaments, liver, blood, cervical tissue, menstrual blood, synovial membranes, skeletal muscle, and even urine [92,93]. In addition to trilineage differentiation potential, MSCs have been well established for their incredible homing potential to the site of injury and regenerative/trophic capacity through paracrine secretion of a myriad of cytokines and growth factors [94,95] and, therefore, they are also being named Medicinal Signaling Cells. Recently, the empirically established promising therapeutic effects attributable to the intrinsic biology of MSCs, in the context of autoimmune diseases, cancer, cardiac/respiratory/skin/kidney disorders, and neurodegenerative disease, have been comprehensively covered by Levy et al. in a critically acclaimed review [91]. In line with this, there has been global approval of several MSC-based therapies, including Temcell HS Inj (JCR Pharmaceuticals, approved in Japan, 2015) for GvHD; Cellgram-AMI (FCB-Pharmicell, approved in South Korea, 2011) for myocardial infarction; Prochymal/remestemcel-L (Osiris, approved in Canada and New Zealand, 2012) for GvHD; Cartistem (Medipost, approved in South Korea, 2012) for knee articular cartilage defect; Neuronata-R (Corestem Inc., approved in South Korea, 2014) for amyotrophic lateral sclerosis; and Alofisel (TiGenix NV/Takeda, approved in Europe, 2018) for complex perianal fistulas in CD, among others [91] (https://alliancerm.org/available-products/, accessed on 25 May 2022). 

MSCs accomplish incredible feat of immunomodulation by interacting with various immune cells through paracrine and/or juxtacrine signaling (cell-to-cell contact dependent mechanism). Among MSC-secreted/-derived cytokines and growth factors mediating paracrine signaling are interleukins (ILs), interferons (IFs), tumor necrosis factors (TNFs), prostaglandins (PGs), vascular endothelial growth factor (VEGF), hepatocyte growth factor (HGF), fibroblast growth factor (FGF), and indoleamine 2, 3-dioxygenase (IDO) [96,97]. For instance, whereas PGE2 secreted by MSCs inhibits functional differentiation of inflammatory T helper 17 cell (Th17) [98], MSC-derived PD-1 ligands potentially interferes with the activation of CD4^+^ T-cells [99], cumulatively suppressing severe inflammatory response. Similarly, MSC-secreted IDO promotes formation of Tregs, which effectively mitigate the immune response [100]. Apart from the paracrine mode of action, MSCs also mediate immune-regulation through direct cellular interaction, involving cells of both innate [natural killer (NK) cells] and adaptive [dendritic cells (DCs), B and T lymphocytes] branches of immune systems [101,102]. The underlying MSC-mediated immunoregulation of immune cells is primarily accomplished via interaction between programmed death molecule (PD-1) with its ligands (PG-L1, PG-L2), IDO-induced inhibition of T cell proliferation and apoptosis, and PGE2-inhibited dendritic cell maturation [103] (Figure 1). This fact seems to be quite relevant and highly promising in the wake of induction of innate and adaptive immune cells following SARS-CoV-2 infection [104], therefore, further exploration of such mechanisms may lead to translatable therapeutic outcome. MSCs are also known to display functional immunomodulatory/immunoregulatory properties both in vitro and in vivo, making them suitable for clinical applications in order to mitigate exacerbated immune response [94], which would otherwise result in severe conditions like ARDS, as observed in critically ill COVID-19 patients [38,41]. A recent finding has shown lack of expression of both ACE2 and TMPRSS2 on human MSCs isolated from both fetal (amnion, cord tissue, cord blood) and adult (adipose tissue and bone marrow) tissues/organs, suggesting MSCs’ nonsusceptibility to SARS-CoV-2 infection. This adds one additional compatible feature in MSCs’ favor as a therapeutic cells for COVID-19 [105]. The inclusion of MSC-based COVID-19 therapy and its clinical trial relies on several well documented evidences derived from both in vitro and in vivo studies. For instance, MSCs facilitate clearance of alveolar fluid as well as restore protein permeability via upregulation of sodium and chloride transporters through paracrine secretion following H5N1 and H7N9 viral infection [106]. In another study involving H9N2-induced lung injury mouse model, single intravenous administration of bone marrow MSCs (BM-MSCs) resulted in considerable reduction in histopathological parameters, such as lung oedema, alveolar tissue injury, and mortality. Such beneficial effects may have resulted from substantial systemic (serum) and localized (bronchoalveolar lavage) reduction in the level of cytokines and chemokines. Compared with BM-MSCs, umbilical cord mesenchymal stem cells (UC-MSCs) have been found to be more effective in terms of restoration of impaired fluid clearance from alveoli, as well as restoration of permeability in an epithelial cell model-based study [107]. The abovementioned beneficial effects of a myriad of tissue-specific MSCs are mediated by paracrine secretion of keratinocyte growth factor, hepatocyte growth factor, and angiopoietin-1 (Ang-1) that collectively act as pro-angiogenic, anti-apoptotic, and pro-regenerative molecules [108]. Briefly, the therapeutic effects, in terms of alleviation of symptoms, reduced inflammation, restored lung function, improved oxygenation, and enhanced survival rate, observed following intravenous infusion of MSCs or MSCs secretome in severe COVID-19 patients is depicted (Figure 2).

## 4. Clinical Trials with Mesenchymal Stem Cells for Severe COVID-19

Mesenchymal stem cells orchestrate both innate and adaptive immune responses and accomplish regenerative potential through the secretion of a range of mediators, such as immunosuppressive molecules, exosomes, chemokines, growth factors, complement components, and various bioactive metabolites. Considering the abovementioned intrinsic biological properties of MSCs, several clinical trials being conducted worldwide to see their effectiveness in severe COVID-19 patients with ARDS (https://www.clinicaltrials.gov/, accessed on 25 May 2022). For instance, the level of inflammatory cytokines, chemokines, and growth factors, such as GM-CSF, IFNγ, IL-5, IL-6, IL-7, TNFα, TNFβ, PDGF-BB, and RANTES, in patients with ARDS, was found to be significantly decreased at day 6 following two intravenous infusions of 100 ± 20 × 10^6^ umbilical cord mesenchymal stem cell (UC-MSCs), one each on day 0 and day 3 [109]. In fact, the beneficial effect of MSCs may be highly durable and last quite a bit longer, as evidenced by improved lung lesion, lower incidence of symptoms, and healthy CT images observed over a 1-year follow up study [110]. In another study, angiopoietin 2, one of the crucial biomarkers of pulmonary and systemic vascular damages [111], has been found to decline significantly following single dose, intravenous administration (over a duration of 60–80 min) of allogeneic bone marrow-derived human mesenchymal stromal cells, highlighting MSC-mediated reversal of such injury [112]. In fact, intravenous infusion of remestemcel-L, human bone marrow-derived MSCs in moderate to severe COVID-19 patients has led to around 83% survival rate, whereas it is only 12% in group that received standard care alone for the same period of treatment, highlighting the incredible therapeutic potential of these stem cells [113].

In addition, secretome derived from stromal cells has been found to significantly increase survival rate, from 28% (control) to 57% (experimental), and other related parameters, apart from leading to considerable decline in ferritin, C-reactive protein, and D-dimer reduction [114]. There are 15 clinical trials reported to have been completed worldwide (https://www.clinicaltrials.gov/, accessed on 25 May 2022), of which some with successful results are mentioned (Table 1), highlighting their prospective potential applications.

## 5. Limitations of MSC-Based Therapeutic Approach

Currently, several animal model-based studies and clinical trials regarding the therapeutic potential of MSCs are being carried out worldwide, and preliminary results in moderately/severally ill patients have shown promising results (Table 1) [116]. Since the first reported MSC-based therapeutic intervention in human subjects by Hillard Lazarus in 1995 [117], MSCs have become the most clinically researched experimental cell for multiple therapeutic purposes, including cancer [118], liver diseases [119], aging [120], sepsis [121], complex infectious diseases [122], and inflammatory diseases [96]. However, there are several limitations of using either MSCs, MSC-EVs, or engineered human MSCs, as explained below. 

The first and foremost limitation entails the lack of large empirical study-based deeper insights into the underlying mechanism of action(s) of MSCs in diseases, such as severe COVID-19 patients, idiopathic pulmonary fibrosis, acute lung injury, and chronic obstructive pulmonary disease (COPD). Owing to the lack of holistic understanding, there is non-availability of standard protocol in terms of dose/number of MSCs (generally, 1–2 million cells/kg), frequency of injection, and spacing thereof. This could be one of the reasons for considerable variability in clinical outcomes reported worldwide. Furthermore, the source of MSC origin and protocols used to culture them before infusion could also result in differential therapeutic effectiveness [123], requiring optimization at the levels of isolation, characterization, and histocompatibility. Moll et al. reported significantly impaired immunomodulation, decreased production of anti-inflammatory mediators, substantial increase in IBMIR (instant blood mediated inflammatory reaction), and a strong activation of the complement cascade following infusion of freeze-thawed MSCs, compared to fresh cells from continuous culture [124], strongly emphasizing the benefits of freshly harvested cells in achieving better clinical outcome. Albeit it imposes new challenges in terms of availability of fresh mesenchymal stem cells in short period of time, especially in case of autologous transplantation.

The second limitation includes a lack of long-term safety evaluations following infusion of MSCs, although short term follow-up studies have not shown considerable adverse effects, barring a few cases. For instance, Xu et al. reported adverse events greater than grade 3 in a small number of patients at one-month follow-up period post MSC infusion [125]. Although, owing to small sample size, the association between reported adverse events (very few) and MSC transplantation remains inconclusive, necessitating large-scale and long-term clinical studies for in-depth safety evaluation.

The third limitation includes prohibitively high cost of cell-based therapy (CBT) as it requires high throughput instruments and an advanced instrumentation facility for isolation, processing, and analysis. After the analysis, culture, and characterization, CBT may also require a storage facility until cells are administered to patients [126,127], enhancing overall cost of treatment. Together, CBT may become very costly, making it unaffordable for the majority of individuals in need, especially to those in low-income category. 

Although clinical administration of MSCs/MSC-EVs to treat COVID-19 and related diseases is still in the preliminary investigation stages (majority at the various phases of the clinical trials) and awaiting wider acceptability among clinicians as a novel therapeutic module, nevertheless, it may be potentially utilized for treating various types of diseases in the near future.

## 6. Conclusions

SARS-CoV-2-induced COVID-19 has adversely influenced and deteriorated various aspects of human existence, including global commerce and economic development, along with causing innumerable debilitations and deaths worldwide. Following the entry of SARS-CoV-2 into the body, innate and adaptive arms of immunity are immediately evoked to resolve infection to prevent disease onset and progression. However, in certain cases, the activated immune response turns potentially pathological rather than protective, thereby causing dreadful and fatal health conditions like acute respiratory distress syndrome (ARDS), and eventual death. Therefore, the collective global fight against COVID-19, based on an array of effective interventions, including drug repurposing [128,129], antibody cocktail [130,131], vaccine [52], and stem cell-based therapy [132], are urgently required to millions of lives. This challenge is increasingly being compounded owing to the origin of various SARS-CoV-2 variants of concerns with enhanced intrinsic pathogenic properties [133]. Among stem cells, MSCs, owing to their intrinsic immunoregulatory properties, have gained considerable prominence as a cell-based therapy to treat ARDS following various types of infections, including SARS-CoV-2, through pre-clinical and clinical trial outcomes. The underlying mechanisms of MSC-based therapy involves regulating inflammatory responses, paracrine secretion of cytokines and growth factors, release of rejuvenating exosomes, and so forth (Figure 1). These well-established therapeutic MSC properties, combined with observed safety and efficacy during pre-clinical and preliminary clinical trials, have been the basis for their large-scale clinical trials (Table 1) to find out the appropriate cell concentration, gap and frequency of dose, and route of administration, in order to develop novel and effective cell-based treatment modalities. Preliminary findings show that MSCs and/or MSC-derived secretome may significantly alleviate respiratory distress, repair lung damage, and turbocharge patient recovery with better immune tolerance [91,112]. Considering all the above-mentioned facts based on empirical research, the unique properties of MSCs may be thoroughly leveraged and used as a potent and versatile therapy in the near future, not only for COVID-19, but also for various other types of disease.

## Figures and Tables

**Figure 1 cells-11-02175-f001:**
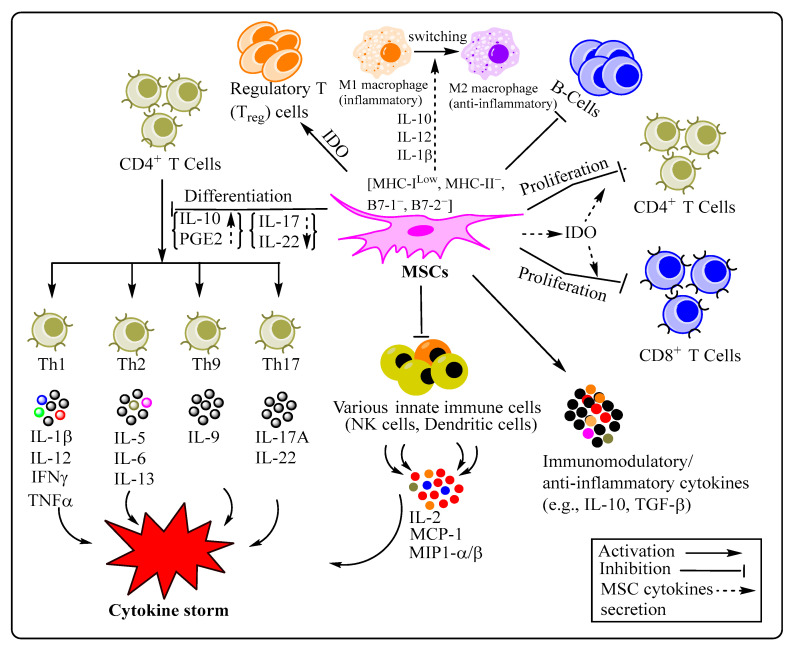
Mesenchymal stem cells (MSCs) accomplish immunomodulation through regulation of various mechanisms (paracrine and juxtacrine signaling) underlying the functioning of diverse innate and adaptive immune cells. Abbreviations: MSCs: Mesenchymal Stem Cells; IL: Interleukin; IFNγ: Interferon gamma; TNFα: Tumor necrosis factor alpha; MCP-1: Monocyte chemoattractant protein-1; MIP1-α/β: Macrophage inflammatory proteins MIP1α (also referred as CCL3) and MIP2α (also referred CXCL2), indoleamine 2, 3-dioxygenase (IDO); PGE2: Prostaglandin E2; TGF-β: Transforming growth factor-β; MHC: Major histocompatibility complex.

**Figure 2 cells-11-02175-f002:**
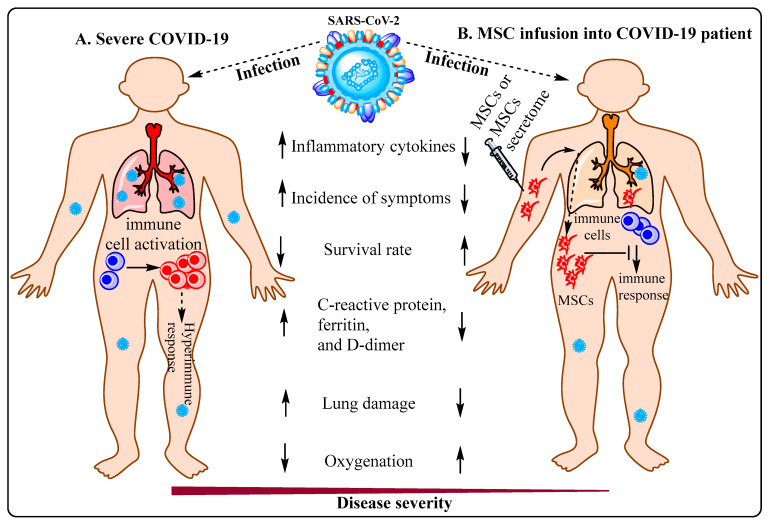
Therapeutic effects of mesenchymal stem cells in severe COVID-19 patients with ARDS. (**A**) Severe COVID-19 patients show enhanced incidence and persistence of symptoms (fever, tiredness, diarrhea, shortness of breath, chest pain, disorientation), increased secretion of inflammatory cytokines and bioactive molecules (C-reactive protein, ferritin, and D-dimer), leading to substantial lung damage and, therefore, hypoxic condition. (**B**) MSCs and/or MSC secretome infusion in severe COVID-19 patients ameliorates disease-specific symptoms, reduce inflammation, restore lung damage and oxygenation, and thereby increases survival rate.

**Table 1 cells-11-02175-t001:** Successful clinical trials involving mesenchymal stem cells in serious/critical COVID-19 patients.

ClinicalTrials.gov Identifier	Phase	Intervention	Clinical Outcomes	Adverse Effects (AEs)	Ref.
Unknown	1/2a	UC-MSC treatment group received two intravenous infusions (at day 0 and 3) of 100 ± 20 × 10^6^ UC-MSCs	Inflammatory cytokines were significantly decreased in UC-MSC-treated subjects at day 6. Treatment led to significantly improved patient survival (91% vs. 42%, *P* = 0.015)	Not significant	[109]
NCT04288102	2	UC-MSCs 3 does of UC-MSCs (4.0 × 10^7^ cells per time) intravenously at Day 0, Day 3, Day 6.	Improvement in whole-lung lesion volume 18% patients showed normal CT image at month 12 compared to none in control Lower incidence of symptoms	Not significant	[110]
NCT02097641	2a	Intravenous, single dose administration (over 60–80 min) of Allogeneic Bone Marrow-Derived Human Mesenchymal Stromal Cells (10 × 10^6^/kg PBW (predicted body weigh)	Significant decrease in angiopoietin 2 in plasma No significant change was detected in the level of IL-6, IL-8, RAGE, or protein C	Not significant	[112]
NCT05019287	1/2	Allogeneic human menstrual blood stem cells (MenSCs) secretome Intravenous injection of 5 mL menstrual blood stem cells secretome on day 1, day 2, day 3, day 4, and day 5	64% of patients had substantial improvement in oxygen levels within 5 days 57% survival rate in treatment group, whereas only 28% in the control group Significant reduction in acute phase reactants, with mean C-reactive protein, ferritin, and D-dimer Significant improvement in lymphopenia	Not significant	[114]
NCT04493242	2	Intravenous administration of bone marrow mesenchymal stem cell-derived extracellular vesicles	Oxygenation improved Significant improvement in absolute neutrophil count and lymphopenia Acute phase reactants, such as C-reactive protein, ferritin, and D-dimer declined	Not significant	[115]
NCT05122234	3	Injection of secretome—mesenchymal stem cell (Single dose of 15 mL dissolved in 100 mL of normal saline, intravenously for 60 min.	The inflamatory markers, including IL-6, IL-10, LIF, VEGF, and Ferritin, are being assessed on day 0 (before intervention), and day 7 and day 14 (after intervention)	Not reported	NA

## Data Availability

Not applicable.

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
