# Peer review of "Exploring the Immunomodulatory Aspect of Mesenchymal Stem Cells for Treatment of Severe Coronavirus Disease 19"

_cells, 2022, doi:10.3390/cells11142175_

Round 1
Reviewer 1 Report
In the present study, although they made many revision, there are some comments for the authors.
1. “2. Immune responses to SARS-CoV-2: Cytokine storm and pathological consequences”, this is not the summarized title. Similarly, “3. MSCs as potential therapeutic cells to mitigate severity of COVID-19 and relevant 362 clinical trials with preliminary findings”. Please also check the sub-title throughout the manuscript.
2. Figure 1. Mesenchymal stem cells (MSCs) accomplish immunomodulation through various mechansims underlying the functioning of diverse innate and adaptive immune cells. Actually, the cytokines are largely missed.
3. Figure 2 should be re-arranged. MSC inhibit immune response rather than immune cells. They are different.
4. Table 1 is not the full information to be presented. Many clinical studies are missed.
Author Response
Comments and Suggestions for Authors
In the present study, although they made many revision, there are some comments for the authors.
Response: First of all, authors would like to express their collective gratitude towards reviewer for invaluable suggestions and comments. Authors have revised the entire manuscript both grammatically and structurally to maintain the seamless flow of information, as well as updated the information wherever needed.
- “2. Immune responses to SARS-CoV-2: Cytokine storm and pathological consequences”, this is not the summarized title. Similarly, “3. MSCs as potential therapeutic cells to mitigate severity of COVID-19 and relevant 362 clinical trials with preliminary findings”. Please also check the sub-title throughout the manuscript.
Response: Titles and subtitles have been revised to make them more inclusive and compatible with the content thereunder.
- Figure 1. Mesenchymal stem cells (MSCs) accomplish immunomodulation through various mechansims underlying the functioning of diverse innate and adaptive immune cells. Actually, the cytokines are largely missed.
Response: Cytokines involved in immunomodulation have been added both in text, especially first half of the introduction section (with proper references) and in figure 1.
- Figure 2 should be re-arranged. MSC inhibit immune response rather than immune cells. They are different.
Response: Figure 2 has been rearranged following the reviewer’s suggestion.
- Table 1 is not the full information to be presented. Many clinical studies are missed.
Response: It’s true that there are several clinical trials relating to MSCs and SAR-CoV-2/COVID-19 are underway (https://www.clinicaltrials.gov/), but most of them are in different phases of completion. Therefore, we have included only those trials which have been completed successfully with some preliminary conclusive findings.
Reviewer 2 Report
The revised manuscript is now well written and the changes have indeed improved it. Especially the section on limitations is appreciated.
Author Response
Comments and Suggestions for Authors
The revised manuscript is now well written and the changes have indeed improved it. Especially the section on limitations is appreciated.
Response: Authors would like to express their collective gratitude towards reviewer for invaluable suggestions and comments, leading to improvement of the manuscript.
Reviewer 3 Report
Dear Authors
I found your paper entitled "Exploring the immunomodulatory aspect of mesenchymal stem cells for treatment of severe coronavirus disease 19" very interesting. The SARS-Cov 2 pandemic has forced researchers around the world to develop new therapies. Your work goes precisely in this direction. I believe it is important not only to develop new conventional drugs but also innovative therapies such as those you have reported in the paper. I think your paper is important to introduce to other researchers the state of the art related to the use of mesenchymal stem cells as a possible therapy for COVID 19. I think your paper is very interesting and may be accepted after minor revision.
minor revision
- line 53-54. it would be desirable to describe all types of mesenchymal stem cells. The one from dental pulp is missing. Please insert relative references (I suggest the papers of Delle Monache et al. 2019 and Arthur A et al. 2008).
- line 57. the authors speak of "tri-lineage" but many papers showed the ability of mesenchymal stem cells to differentiate toward the neuronal lineage (Rafiee F et al 2020 and Martellucci S et al 2019).
- line 114. I suggest the authors introduce the SARS-CoV 2 with a short structural and taxonomic description (you can expand the written part in the abstract line 18-19) and related references (I suggest Sorice M et al 2021 and ICT virus taxonomy)
- references section. Modify the references according to the instructions for the authors:
Author 1, A.B.; Author 2, C.D. Title of the article. Abbreviated Journal Name Year, Volume, page range. Furthermore, within the text, the references have to be inserted inside square brackets.
- line 389. Correct the citations in (86-88).
- line 457. correct the word "thses" in "these".
- line 480. correct the word "standarad" to "standard".
Author Response
Comments and Suggestions for Authors
Dear Authors
I found your paper entitled "Exploring the immunomodulatory aspect of mesenchymal stem cells for treatment of severe coronavirus disease 19" very interesting. The SARS-Cov 2 pandemic has forced researchers around the world to develop new therapies. Your work goes precisely in this direction. I believe it is important not only to develop new conventional drugs but also innovative therapies such as those you have reported in the paper. I think your paper is important to introduce to other researchers the state of the art related to the use of mesenchymal stem cells as a possible therapy for COVID 19. I think your paper is very interesting and may be accepted after minor revision.
Response: First of all, authors would like to express their collective gratitude towards reviewer for invaluable suggestions and comments. Authors have revised the entire manuscript both grammatically and structurally to maintain the seamless flow of information, as well as updated the information wherever needed.
minor revision
- line 53-54. it would be desirable to describe all types of mesenchymal stem cells. The one from dental pulp is missing. Please insert relative references (I suggest the papers of Delle Monache et al. 2019 and Arthur A et al. 2008).
Response: Incorporated as per suggestion.
- line 57. the authors speak of "tri-lineage" but many papers showed the ability of mesenchymal stem cells to differentiate toward the neuronal lineage (Rafiee F et al 2020 and Martellucci S et al 2019).
Response: Incorporated as per suggestion.
- line 114. I suggest the authors introduce the SARS-CoV 2 with a short structural and taxonomic description (you can expand the written part in the abstract line 18-19) and related references (I suggest Sorice M et al 2021 and ICT virus taxonomy)
Response: Incorporated with elaboration as per suggestion.
- references section. Modify the references according to the instructions for the authors:
Response: Authors highly appreciate the suggestions regarding in-text and bibliography. All the references have been double-checked with the insertion of few new references as per the content addition.
Author 1, A.B.; Author 2, C.D. Title of the article. Abbreviated Journal Name Year, Volume, page range. Furthermore, within the text, the references have to be inserted inside square brackets.
Response: Done as per suggestion.
- line 389. Correct the citations in (86-88).
Response: Corrected
- line 457. correct the word "thses" in "these".
Response: Corrected
- line 480. correct the word "standarad" to "standard".
Response: Corrected
This manuscript is a resubmission of an earlier submission. The following is a list of the peer review reports and author responses from that submission.
Round 1
Reviewer 1 Report
This review article needs a lot of improvement.
- There are many grammatical and typo errors.
- It reads like a literature summary rather than reviewing the findings, lacking the authors' perspectives and critics of the published works related to MSCs.
- The authors failed to describe the limitation of the MSC-based therapeutic approach.
Reviewer 2 Report
In the present study, Chaudhary et al. summarized MSC for treating COVID-19. Although it is interesting, there are some comments for the authors.
- Title is not common to use, “Mesenchymal Stem Cell-based therapeutic Immunomodulation for treatment of severe Coronavirus disease 19 (COVID-19)”, which is wrong expression. Please revise the language.
- “After an average incubation period of 2-14 days, the majority of infected individuals remain asymptomatic”, this is not true.
- Page 4, Line 112, IL-10 is pro-inflammatory cytokine? Figure 1 has indicated that IL-10 is anti-inflammatory cytokine. This is self-contradictory. “The Role of the Anti-Inflammatory Cytokine Interleukin-10 in Tissue Fibrosis” has clear explanation for the role of IL-10.
- How to confirm the role of innate immunity or adaptive immunity, from the description, these cytokines can be released from many kinds of cells rather than immune cells.
- There are many clinical studies are lacked for authors, please supply more information. And there are two reviews (PMID: 35279630 and PMID: 35187617) can be suggested.
- “Administration of MSCs as therapeutic cells to mitigate cytokine storm culminating 330 in ARDS”, this is not related to the subject.
- The challenge of MSC and COVID-19 therapy should be added.
- Figure 2 is not clear for the description.
- What is the standard for severe COVID-19, the structure should be adjusted.